# CAPL: Graph Few-Shot Class-Incremental Learning Via Class-Adaptive Prototype Learning

## Abstract

Few-shot class-incremental learning has always been a challenging problem due to catastrophic forgetting, insufficient labels and class imbalance. Graph few-shot class-incremental learning(GFSCIL), with the presence of edges between nodes and complex relationships between classes, further increases the difficulty of the learning process. Current researches in this field mainly employ meta-learning and metric-learning approaches. However, these methods do not consider the relationships between classes and treat all classes equally, which does not conform to the real-world applications. To address these limitations, we propose a class-adaptive prototype learning (CAPL) method that adaptively processes each class based on the relationships between classes, thereby alleviating spatial confusion between new and old classes as well as the catastrophic forgetting problem. Specifically, we first adopt a class-adaptive spatial reservation module to allocate larger spaces for the arrival of new classes, preventing confusion between new and old classes. We then utilize a class-adaptive prototype alignment module for knowledge distillation. By considering the positional relationship between new and old classes in the feature space, we provide greater flexibility to classes closely related to new classes while retaining classification information of old classes, thus adapting to the arrival of new classes. Experiment results demonstrate the superiority of the proposed method.

## 1 Introduction

Incremental Learning (IL) serves as an advanced learning paradigm, enabling models to progressively acquire and incorporate knowledge from novel data scenarios while simultaneously maintaining proficiency in tasks learned previously Cauwenberghs & Poggio (2000); Kuzborskij et al. (2013); Mensink et al. (2013). A key assumption in this process traditionally lies in the requirement for abundant annotation data tied to new tasks, which is leveraged to re-train or fine-tune the model Rebuffi et al. (2017). Unfortunately, sourcing large numbers of samples relevant to these new tasks remains an unrealistic expectation in the context of real applications.

In contrast, the human learning process exhibits an ability to not only grasp new concepts from minimal examples but also to retain previously learned knowledge concurrently. It is, therefore, an attractive prospect to devise algorithms aiding in the incremental learning process with sparse samples. Presently, academic probing into few-shot class-incremental learning predominantly targets image-based Chen & Lee (2020); Deng & Xiang (2024); Rebuffi et al. (2017); Zhou et al. (2022a) and natural language processing domainsQin & Joty (2021; 2022); Wang et al. (2022). The exploration of this learning paradigm within the graph data has been comparatively overlooked. To our current knowledge, a mere two works, namely HAG-MetaTan et al. (2022) and GeometerLu et al. (2022), have tackled this matter, employing meta-learning and metric-learning techniques, respectively.

The task of graph few-shot class-incremental learning is confronted by four principal challenges. First, during the assimilation of new class-specific data, models tend to relinquish their stored knowledge from the existing classes, i.e., catastrophic forgetting Goodfellow et al. (2013); Kirkpatrick et al. (2017); Yap et al. (2021), referring to the decrement in performance when processing previ-

ously learnt classes in the face of incremental new ones. The second hurdle pertains to the deficit in predictive accuracy on newly introduced classes, typically a repercussion of overfitting on limited training samples of new classes Ding et al. (2020); Peng et al. (2022); Snell et al. (2017). The third constraint could be attributed to feature intersection between new and old classes, resulting in class ambiguity within the feature space Chen & Lee (2020); Deng & Xiang (2024); Zhou et al. (2022a). Lastly, the non-identically distributed manner of graph nodes further complexifies the process of graph incremental learning. Graph neural networks employ edges as conduits for inter-nodal information transmission, catalyzing differentiated interrelationships among nodes of disparate classes Cao et al. (2016); Defferrard et al. (2016); Hamilton et al. (2017); Veličković et al. (2017); Xu et al. (2018). The development of unique processing mechanisms for distinct classes based on these class interrelationships stands as an integral, albeit frequently disregarded, challenge Lu et al. (2022); Tan et al. (2022) .

To address the aforementioned challenges, we put forth an novel Class-Adaptive Prototype Learning (termed CAPL) method for graph few-shot class-incremental learning. The essence of CAPL lies in its capacity to compute a distinct prototype representation for each class, acting as the class representative. It then performs class categorization based on the distance between nodes and their corresponding prototypes, thereby curtailing the risk of potential overfitting sparked by sample scarcity.

Specifically, to stave off catastrophic forgetting during the incorporation of new class, in CAPL, we harness Class-Adaptive Prototype Alignment (CAPA) based knowledge distillation to preserve the classification integrity of legacy classes. CAPA, standing in contrast to previous methodologies Dong et al. (2021); Gou et al. (2021); Hinton et al. (2015); Lu et al. (2022) which upheld uniformity across all classes during knowledge distillation, takes into account the intricate relationships binding old and new classes. For old classes that are susceptible to confusion with new classes within the feature landscape, we confer upon them the liberty of increased flexibility during the knowledge distillation phase. This effectively aids in averting class overlap. Concretely, if an old class exhibits proximity to several new classes, it is endowed with enhanced flexibility. Further, we find that in earlier studies Chen & Lee (2020); Deng & Xiang (2024); Zhou et al. (2022a), the approach of retaining space during the training phase to accommodate imminent new classes was suggested, thereby mitigating confusion between old and new classes, and in turn bolstering classification precision. However, these methods indiscriminately treat all classes, rendering them potentially ill-suited for graph few-shot class-incremental learning due to the convoluted edge relationships interconnecting nodes. Therefore, we introduce the Class-Adaptive Prototype Clustering (CAPC) module further to embrace differential space reservation strategies rooted in the relationships woven between classes. This class-specific adaptation, in turn, enhences the model's classification capabilities to a respective degree.

Our contribution can be summarised as follows:

- We investigated few-shot class-incremental learning on graph data, and for the first time, proposed using a class-adaptive approach to handle different classes differently during the incremental learning process to avoid confusion between classes.

- We have proposed a novel class-adaptive model CAPL to address the GFSCIL problem, employing distinct space reservation and knowledge distillation strategies for different classes. To the best of our knowledge, CAPL is the first model in class-incremental learning to handle the diversity among different classes based on their relationships.

- The experimental results on three commonly used graph datasets Cora-Full, Amazon-Electronics, and Amazon-Clothing demonstrate that our method surpasses state-of-the-art performance. Particularly, our method achieves an average accuracy improvement of over 3% on datasets Cora-Full and Amazon-Electronics compared to the closest strong baseline.

## 2 PROBLEM STATEMENT

In this section, we present the problem statement and definitions. In graph incremental learning problem, we begin with a base graph $\mathcal{G}_0 = \{A_0, X_0, Y_0\}$ where each class initially possesses abundant labeled samples. In the streaming sessions, suppose we have $T$ snapshots of evolving graph, denoting as $\mathcal{G}^{stream} = \{\mathcal{G}_1, \ldots, \mathcal{G}_T\}$, where $\mathcal{G}_t = \{A_t, X_t, Y_t\}$. Take the $t$-th session as an example, suppose we have $N_t$ nodes and $M_t$ edges, $X_t = \{x_1, \ldots, x_{N_t}\} \in \mathbb{R}^{N_t \times d}$ is the node feature

matrix, $A_t \in \mathbb{R}^{N_t \times N_t}$ denotes the adjacency matrix. We denote $\{\mathcal{C}_0, \mathcal{C}_1, \cdots, \mathcal{C}_T\}$ as sets of classes from base stage to the $T$-th streaming session of incremental learning stage. $\mathcal{C}_0$ is the set of base classes with plenty of labeled samples and $Y_0 \in \mathcal{C}_0$. In $t$-th ($t > 0$) session, $Y_t \in \mathcal{C}_t$, and $\Delta\mathcal{C}_t$ new classes are introduced with few-shot samples, where $\forall i, j, \Delta\mathcal{C}_i \cap \Delta\mathcal{C}_j = $ and $\mathcal{C}_i = \mathcal{C}_{i-1} + \Delta\mathcal{C}_i$. Thus, the total encountered classes in $t$-th session can be denoted as $\mathcal{C}_t = \mathcal{C}_0 + \sum_{i=1}^{t} \Delta\mathcal{C}_i$.

**Problem 1. Graph few-shot class-incremental learning.** In $t$-th streaming session, we denote $\Delta\mathcal{C}_t$ novel classes with $K$ labeled nodes as the $\Delta\mathcal{C}_t$-way $K$-shot graph few-shot class-incremental learning problem. The labeled training samples are denoted as support sets $S_t$. Another batch of nodes to predict their corresponding label are denoted as query sets $Q_t$. After training on the support sets $S_t$, the GFSCIL problem is tested to classify unlabeled nodes of query sets $Q_t$ into all encountered classes $\mathcal{C}_t$. It is important to note that in the base stage, both the support set and the query set are used in learning the model parameters, while in the incremental learning stage, the support set is used in training while the query set is used in testing.

**Definition 1. Prototype Representation.** The prototype representation is used to characterize a typical embedding of a class. Generally, the node embeddings of a class tend to cluster around its prototype representation in the same metric space. This concept was first introduced in Snell et al. (2017), where the mean of all instances in the support set is used as the prototype representation of the class.

## 3 METHODOLOGY

Besides the complex connections between samples, there are also complicated relationships between different classes, exploring deeply on the connections within classes would greatly benefit the continual learning of novel classes, especially in few-shot learning scenarios. Considering this, firstly, to accommodate new classes that may be introduced in subsequent phases, we prioritize enhancing intra-class contraction and inter-class separation for classes that are particularly susceptible to boundary confusion. This ensures that there is ample space for the inclusion of new classes and reduces the risk of misclassification of new-class samples. Secondly, we propose a novel knowledge distillation method to effectively preserve the classification information of samples from existing classes by aligning class prototypes that have a lower likelihood of confusion with the prototypes present in the old model. By doing so, we ensure that the learned knowledge from the old model is transferred and integrated seamlessly into the updated model, thereby maintaining the integrity and consistency of the classification process. Thus, we can conduct a class-adaptive processing with the considering of relationships between classes and achieve better few-shot class continuous learning. The overall framework is illustrated in Figure 1.

### 3.1 NODE CLASSIFICATION VIA PROTOTYPE LEARNING

**Base stage.** Prototype learning studies a clear and tight representation for each class to capture the essential characteristics of the classes, and classifies samples based on their proximity to class-specific prototypes. Specifically, for a graph $\mathcal{G} = \{A, X\}$, we first use a GNN encoder $g_\theta$ to aggregate node information and map it into a vector space where the latent features can be denoted as:

$$\boldsymbol{Z} = g_\theta(A, X) \tag{1}$$

Then, for class $i$, the simplest way to obtain a prototype is to average the representations of the nodes in class $i$, i.e., $p_i = \frac{1}{|\mathcal{C}_i|} \sum_{j \in \mathcal{C}_i} z_i$, where $|\mathcal{C}_i|$ is the number of nodes in class $i$. However, we find that this approach is susceptible to noise. To mitigate this effect, we take into account both the similarity between nodes of the same class and the importance of nodes within the graph. Thus, we introduce another attention network $g_\varphi$. for class $i$, to process the features generated by $g_\theta$:$\boldsymbol{Z}_i^{spt} = \text{CONCATENATE}(\|_{v_j \in S_t^i} \boldsymbol{z}_j)$, represent the support node representations of class $i$, making it more attentive to critical data, i.e.,

$$\overline{\boldsymbol{Z}}_i^{spt} = g_\varphi(\boldsymbol{Q}, \boldsymbol{K}, \boldsymbol{V}) = \text{softmax}(\frac{\boldsymbol{Q}\boldsymbol{K}^T}{\sqrt{d_k}})\boldsymbol{V} + \boldsymbol{Z}_i^{spt} \tag{2}$$

where $\boldsymbol{Q} = \mathbf{W}_Q \boldsymbol{Z}_i^{spt}, \boldsymbol{K} = \mathbf{W}_K \boldsymbol{Z}_i^{spt}, \boldsymbol{V} = \mathbf{W}_V \boldsymbol{Z}_i^{spt}$, $\mathbf{W}_Q, \mathbf{W}_K, \mathbf{W}_V$ are three parameter matrices and $d_k$ is the dimension of $\boldsymbol{Q}$ and $\boldsymbol{K}$. We then use node degrees as weights in the prototype

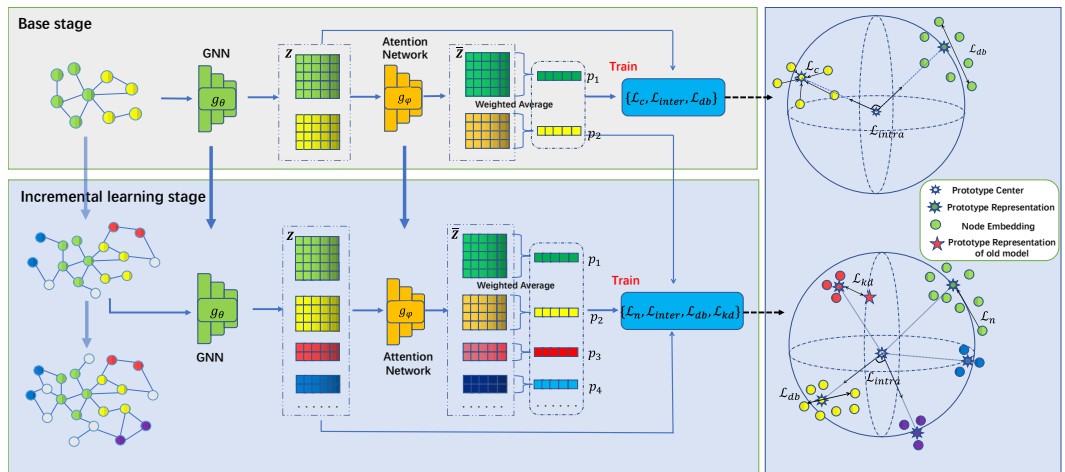

Figure 1: The overview of proposed method for GFSCIL. The left part illustrates the overall training process of the CAPL model in graph few-shot class incremental learning. The right part explains from a geometric perspective the roles of several loss functions proposed by CAPL.

calculation, applying a weighted average to the transformed features:

$$\boldsymbol{p}_i = \sum_{j \in \boldsymbol{C}_i} \frac{\text{degree}(v_j)}{\sum_{j' \in \boldsymbol{C}_i} \text{degree}(v_{j'})} \cdot \overline{\boldsymbol{z}}_j, \tag{3}$$

Then we use the prototypes for classification of nodes. Intuitively, a node should be close to prototype of its class. Thus, we design the following loss function $\mathcal{L}_c$ to encourage nodes to cluster around their prototypes, i.e.,

$$\mathcal{L}_c = \frac{1}{N} \sum_{i=1}^{\|\mathcal{C}_0\|} \sum_{k=1}^{n_i} -\log \frac{\exp\left(-\|\boldsymbol{z}_k - \boldsymbol{p}_i\|^2\right)}{\sum_{i' \in \mathcal{C}_i} \exp\left(-\|\boldsymbol{z}_k - \boldsymbol{p}_{i'}\|^2\right)}, \tag{4}$$

where $\|\mathcal{C}_0\|$ is the number of class in base stage, and $n_i$ represents the node samples of class $i$.

**Incremental learning stage.** In this stage, the model learns and adapts to new classes, and tries to avoid the potential risk of forgetting old class information. A new incremental classification loss $\mathcal{L}_n$ is proposed to achieve this goal, with selectively updating model's parameters only when the prototype closest to the node's embedding is not the prototype of its true class. The specific loss function is as follows:

$$\mathcal{L}_n = \frac{1}{N} \sum_{k=1}^{\|\mathcal{C}_t\|} \sum_{i=1}^{n_k} \frac{\|\boldsymbol{z}_i - \boldsymbol{p}_k\|^2 - \min_{j \in \{\mathcal{C}_t\}} \|\boldsymbol{z}_i - \boldsymbol{p}_j\|^2}{\|\boldsymbol{z}_i - \boldsymbol{p}_k\|^2 + \min_{j \in \{\mathcal{C}_t\}} \|\boldsymbol{z}_i - \boldsymbol{p}_j\|^2} \tag{5}$$

For node $v_i$, when $\min_{j \in \{\mathcal{C}_t\}} \|\boldsymbol{z}_i - \boldsymbol{p}_j\|^2 = \|\boldsymbol{z}_i - \boldsymbol{p}_k\|^2$, the node can be correctly classified. In this situation, the loss function equals 0, and no parameter updates are performed.Otherwise, if $\min_{j \in \{\mathcal{C}_t\}} \|\boldsymbol{z}_i - \boldsymbol{p}_j\|^2 < \|\boldsymbol{z}_i - \boldsymbol{p}_k\|^2$, indicating that the sample is misclassified, we update the parameters accordingly. We also use the denominator to scale the data to the range of [-1, 1]. This helps to reduce the frequency of parameter updates while bringing the node embedding closer to the prototypes, thereby avoiding the forgetting of classification information related to old classes.

Furthermore, to further enhance the discriminability between prototypes belonging to different classes and consequently improve the overall classification accuracy, we introduce a novel loss function termed as the inter-class prototype similarity loss. The formulation of this loss function, denoted as $\mathcal{L}_{inter}$, is presented as follows:

$$\mathcal{L}_{inter} = -\frac{1}{\|\mathcal{C}_k\|} \sum_{i=1}^{\|\mathcal{C}_k\|} \log \left[ 1 - \max_{j \in \{\mathcal{C}_k\} \setminus i} \frac{\overline{\boldsymbol{p}}_i}{\|\overline{\boldsymbol{p}}_i\|} \cdot \frac{\overline{\boldsymbol{p}}_j}{\|\overline{\boldsymbol{p}}_j\|} \right] \tag{6}$$

where $\overline{\boldsymbol{p}}_i = \boldsymbol{p}_i - \frac{1}{\|\mathcal{C}^k\|} \sum_{j=1}^{\|\mathcal{C}^k\|} \boldsymbol{p}_j$, and $\frac{\overline{\boldsymbol{p}}_i}{\|\overline{\boldsymbol{p}}_i\|}$ projects all nodes onto the unit sphere. The purpose of using logarithmic loss and maximum values is to make the model pay more attention to the closer prototype representations.

### 3.2 CLASS-ADAPTIVE PROTOTYPE CLUSTERING

The adopted losses $\mathcal{L}_n$ and $\mathcal{L}_{inter}$ would encouraging the separability of distances between nodes and prototypes, as well as the angles between prototypes. We need further improve the compactness of nodes in the same classes to reserve more spare space for new classes and to reduce feature overlap between new classes and existing ones. To achieve this, different with previous works Chen & Lee (2020); Deng & Xiang (2024); Zhou et al. (2022a) which treat existing classes equally and ignore the connection between classes, we explore class similarities with the help of graph structure and pay more attention on separation of adjacent classes and similar classes.

Specifically, we introduce Davies-Bouldin Index Davies & Bouldin (1979), a metric that can assess quality of clustering by measuring the compactness of distances between samples within the same cluster and the separability of distances between different cluster centers. We employ this index as a loss function to encourage closer aggregation of samples within the same class, thereby achieving supervised prototype clustering. To our knowledge, this is the first time that a clustering evaluation metric has been applied to the problem of few-shot class incremental learning. The specific loss function is as follows:

$$\mathcal{L}_{db} = \frac{1}{\|\mathcal{C}_t\|} \sum_{k=1}^{\|\mathcal{C}_t\|} \max_{j \neq k} \frac{s_k + s_j}{d_{kj}} \tag{7}$$

$$s_k = \frac{1}{n_k} \sum_{i=1}^{n_k} \|\boldsymbol{z}_i - \boldsymbol{p}_k\|^2, \, d_{kj} = \|\boldsymbol{p}_k - \boldsymbol{p}_j\|^2 \tag{8}$$

Where $s_k$ and $d_{kj}$ measure the compactness of intra-class samples and the separability of inter-class prototypes, respectively. Specifically, during the process of reserving space for new classes, $\mathcal{L}_{db}$ focus differently on various classes based on the relationships between them. On one hand, for classes that are close in the feature space and more prone to boundary confusion, we aim to maintain larger spaces between them. Thus, we use distance as a weight to encourage better separation between closely situated classes. On the other hand, when a class is close to multiple classes, we use the $max$ function to focus more attention on the intermediary class. This approach simultaneously expands the space between the intermediary class and the surrounding classes, thereby reserving more space for subsequently added classes.

### 3.3 CLASS-ADAPTIVE PROTOTYPE ALIGNMENT

Further, to alleviate catastrophic forgetting during the incremental learning stage, a new knowledge distillation method is introduced, i.e., the Class-Adaptive Prototype Alignment(CAPA). Unlike previous knowledge distillation methods Dong et al. (2021); Gou et al. (2021); Hinton et al. (2015); Lu et al. (2022) that apply same strategy to all classes, CAPA assigns class-specific weights to old classes and take the connections between them and new classes into consideration. For classes that are closely related to new classes, CAPA provides more flexibility by performing prototype alignment with smaller weights, enabling them to better differentiate themselves from the new classes. We define the set $\{c_{i,1}, c_{i,2}, \cdots, c_{i,|\mathcal{C}_t|}\} = \mathcal{C}_{t-1}$ such that the distances satisfy: $d(c_{i,1}, c_i) < d(c_{i,2}, c_i) < \cdots < d(c_{i,|\mathcal{C}_{t-1}|}, c_i)$. For a new class $c_i \in \mathcal{C}_t$, we assign a weight $a_{ij}$ to old class $c_j \in \mathcal{C}_{t-1}$:

$$a_{ij} = \begin{cases} \alpha, & if \ c_j \in \{c_{i,1}, c_{i,2}, \cdots, c_{i,k}\} \\ 1, & if \ c_j \in \{c_{i,k+1}, c_{i,k+2}, \cdots, c_{i,|\mathcal{C}_{t-1}|}\} \end{cases} \tag{9}$$

where $1 > \alpha > 0, k > 0$ are hyper-parameters, and the inter-class distance is obtained by calculating the Euclidean distance between class prototypes. It is worth noting that since our goal is to align the prototypes between the teacher model and the student model, the computed distance is between the prototypes of the old classes in the teacher model and the new classes in the student model. For old

| Datasets | Cora-Full | Amazon-Clothing | Amazon-Electronics |
|----------|-----------|-----------------|--------------------|
| Nodes    | 19, 793   | 24, 919         | 42, 318            |
| Edges    | 126, 842  | 91, 680         | 43, 556            |
| Features | 8, 710    | 9, 034          | 8, 669             |
| Class    | 70        | 77              | 167                |

Table 1: The statistics of datasets.

class $c_j \in \mathcal{C}_{t-1}$, the corresponding weight $a_j$ is defined as:

$$a_j = \prod_{i=1}^{\|\Delta \mathcal{C}_t\|} a_{ij} \tag{10}$$

where $\|\Delta \mathcal{C}_t\|$ is the number of new classes in session $t$. Then, using these weights, the prototype representations are adapted through class-adaptive alignment. With these weights, we perform class-adaptive alignment on the prototypes of the old classes. The specific knowledge distillation loss function $\mathcal{L}_{kd}$ is as follows:

$$\mathcal{L}_{kd} = \frac{1}{\|\mathcal{C}_{t-1}\|} \sum_{j=1}^{\|\mathcal{C}_{t-1}\|} a_j \cdot \|\boldsymbol{p}_j - \boldsymbol{p}'_j\|^2 \tag{11}$$

where $\boldsymbol{p}'_j$ is the prototype representation of class $j$ in the old model. When the prototype distance between an old class and a new class is large, they are less likely to overlap. We aim to preserve the classification information of that old class by maintaining its prototype in its original position, therefore assigning a greater weight to its prototype alignment. In contrast, when an old class is close to the prototype of a new class, we assign a smaller weight to allow more flexibility for the old class prototype to adjust, which enhances classification performance.Notably, when more new classes are closer to a particular old class, we assign greater flexibility to that old class.

In summary, the overall loss function for the base stage is as follows:

$$\mathcal{L}_{base} = \mathcal{L}_c + \lambda_1 \mathcal{L}_{inter} + \lambda_2 \mathcal{L}_{db} \tag{12}$$

where $\lambda_1$ and $\lambda_2$ are hyper-parameters. And the total loss function for the incremental learning stage $\mathcal{L}_{IL}$ is given by:

$$\mathcal{L}_{IL} = \mathcal{L}_n + \lambda_3 \left( \mathcal{L}_{inter} + \mathcal{L}_{db} + \mathcal{L}_{kd} \right) \tag{13}$$

where $\lambda_3$ is hyper-parameter.

## 4 EXPERIMENT

### 4.1 EXPERIMENTAL SETUP

**Datasets.**We evaluate the performance of the model on three commonly used real-world graph datasets: Cora-Full, Amazon-Clothing and Amazon-Electronics. The statistics are shown in Table 1. A detailed description of these three datasets is provided in Appendix B.1.

**Baselines.** To validate the effectiveness of the proposed CAPL method, the following baselines are included in the experiments: PNSnell et al. (2017), GPNDing et al. (2020), ER-GNNZhou & Cao (2021), GeometerLu et al. (2022), HAG-MetaTan et al. (2022), IDLVQChen & Lee (2020), FACTZhou et al. (2022a), EHSDeng & Xiang (2024), TEENWang et al. (2024). Note that before feeding graph data into models that are not equipped to process such data, the data are first embedded by a two-layer GAT. A detailed introduction is provided in the Appendix B.2.

**Dataset split.** In the experiments, we divided the datasets into data for base phase and data for multiple incremental learning phases respectively. For the Cora-Full dataset, we randomly select 20 classes for the base phase, and the rest 50 classes are divided into 10 part for the incremental learning phases, following the 5-way-5-shot setting. Towards dataset Amazon-Clothing, following the setting of HAG-Meta Tan et al. (2022), we selected 50 classes as the base classes, and the rest 27 classes are divided equally to the 9 incremental learning phases, and follows the 3-way-5-shot

setting. Towards the Amazon-Electronics dataset, we selected 67 classes as the base classes, and the rest 100 classes are remained for the 10 incremental learning stages, divided in the 10-way-5-shot setting.

**Hyperparameter setting.** For a fair comparison, all methods use a two-layer GATVeličković et al. (2017) with 512 hidden neurons. And for methods on images, the encoder is replaced with a GAT, and the rest remains unchanged. The learning rate is 1e-3 during the base phase and 1e-4 during the incremental phases. The number of neighbors $k$ is 5, and the decay coefficient $\alpha$ is 0.4.

**Evaluation Metrics.** We adopt two evaluation metrics to assess the model's performance: average forgetting and average accuracy. Average forgetting quantifies the extent of catastrophic forgetting in the model, with a lower value indicating a smaller loss of classification information. The corresponding formula is: $AF = \frac{1}{T}\sum_{t=1}^{T} Acc(t) - Acc(t-1)$, where $Acc(t)$ is the classification accuracy of session $t$. Average accuracy represents the mean classification accuracy across all sessions, where higher values signify better overall classification performance.

| Methods | Cora-Full dataset (5-way 5-shot) | | | | | | | | | | | Avg. Forgetting | Avg. Acc. | Improv-* ement |
|---|---|---|---|---|---|---|---|---|---|---|---|---|---|---|
| | 0 | 1 | 2 | 3 | 4 | 5 | 6 | 7 | 8 | 9 | 10 | | | |
| PN | 80.45 | 59.80 | 52.11 | 44.04 | 41.25 | 36.94 | 35.09 | 34.20 | 32.11 | 30.51 | 29.54 | 5.09 | 43.27 | **+13.43** |
| GPN | **81.51** | 61.21 | 54.30 | 45.42 | 41.44 | 37.04 | 35.18 | 34.30 | 33.66 | 32.79 | 32.15 | 4.94 | 44.45 | **+10.82** |
| ER-GNN | 80.28 | 60.29 | 52.58 | 45.20 | 43.25 | 39.44 | 39.53 | 37.62 | 36.43 | 37.02 | 36.68 | 4.36 | 46.21 | **+6.29** |
| IDLVQ | 79.81 | 62.91 | 56.21 | 47.33 | 44.72 | 41.22 | 39.93 | 38.58 | 36.65 | 35.88 | 35.97 | 4.38 | 47.20 | **+7.00** |
| FACT | 80.26 | 64.87 | 55.07 | 45.99 | 42.23 | 40.89 | 40.78 | 38.96 | 38.27 | 37.98 | 36.83 | 4.34 | 47.46 | **+6.14** |
| EHS | 76.67 | 64.51 | 56.37 | 47.31 | 43.39 | 40.51 | 39.30 | 39.17 | 38.05 | 36.96 | 36.49 | 4.02 | 47.15 | **+6.48** |
| TEEN | 81.15 | 62.05 | 54.04 | 47.93 | 44.93 | 40.91 | 40.56 | 39.76 | 38.57 | 39.72 | 38.37 | 4.28 | 47.99 | **+4.60** |
| HAG-Meta | 79.16 | 68.91 | 58.28 | 52.36 | 47.68 | 43.59 | 40.78 | 38.43 | 37.91 | 34.12 | 35.15 | 4.40 | 48.76 | **+7.82** |
| Geometer | 79.68 | 69.44 | 61.26 | 54.59 | 47.88 | 44.14 | 41.93 | 42.04 | 40.83 | 40.24 | 39.60 | 4.01 | 51.05 | **+3.37** |
| **CAPL(ours)** | 80.29 | **71.17** | **64.47** | **56.62** | **52.54** | **48.88** | **47.41** | **46.54** | **44.92** | **43.52** | **42.97** | **3.73** | **54.51** | |

| Methods | Amazon-Electronics dataset (10-way 5-shot) | | | | | | | | | | | Avg. Forgetting | Avg. Acc. | Improv-* ement |
|---|---|---|---|---|---|---|---|---|---|---|---|---|---|---|
| | 0 | 1 | 2 | 3 | 4 | 5 | 6 | 7 | 8 | 9 | 10 | | | |
| PN | 75.23 | 67.79 | 62.33 | 59.42 | 55.73 | 51.70 | 49.92 | 46.83 | 45.14 | 44.02 | 42.87 | 3.24 | 54.63 | **+16.19** |
| GPN | 75.03 | 68.19 | 64.02 | 62.75 | 60.08 | 56.66 | 55.36 | 53.04 | 51.24 | 50.19 | 49.28 | 2.58 | 58.71 | **+9.78** |
| ER-GNN | 75.05 | 67.37 | 61.31 | 59.31 | 55.06 | 50.92 | 49.60 | 47.95 | 46.06 | 44.76 | 43.68 | 3.14 | 54.64 | **+15.38** |
| IDLVQ | 74.65 | 70.81 | 66.93 | 65.61 | 63.03 | 59.43 | 57.75 | 55.94 | 53.63 | 52.63 | 50.98 | 2.37 | 61.03 | **+8.08** |
| FACT | **77.00** | **72.76** | 69.21 | 66.97 | 63.37 | 59.80 | 59.37 | 57.12 | 56.00 | 54.85 | 54.38 | 2.26 | 62.8 | **+4.68** |
| EHS | 73.62 | 68.59 | 65.63 | 63.83 | 61.32 | 57.66 | 56.34 | 54.47 | 52.60 | 51.20 | 51.02 | 2.26 | 59.66 | **+8.04** |
| TEEN | 76.72 | 70.06 | 64.42 | 63.21 | 58.86 | 54.00 | 52.34 | 49.55 | 48.34 | 47.30 | 46.51 | 3.02 | 57.39 | **+12.55** |
| HAG-Meta | 74.12 | 72.12 | 66.52 | 64.42 | 61.74 | 58.85 | 57.94 | 56.49 | 55.23 | 55.68 | 54.28 | 1.98 | 61.58 | **+4.78** |
| Geometer | 75.24 | 70.06 | 65.34 | 64.02 | 61.58 | 59.43 | 57.19 | 54.33 | 52.88 | 51.07 | 50.16 | 2.51 | 60.11 | **+8.90** |
| **CAPL(ours)** | 75.66 | 72.50 | **70.51** | **70.50** | **68.50** | **64.34** | **62.91** | **61.35** | **60.01** | **59.82** | **59.06** | **1.66** | **65.46** | |

| Methods | Amazon-Cloting dataset (3-way 5-shot) | | | | | | | | | | Avg. Forgetting | Avg. Acc. | Improv-* ement |
|---|---|---|---|---|---|---|---|---|---|---|---|---|---|
| | 0 | 1 | 2 | 3 | 4 | 5 | 6 | 7 | 8 | 9 | | | |
| PN | 75.00 | 72.15 | 71.32 | 68.10 | 66.42 | 58.13 | 55.92 | 52.68 | 50.53 | 49.49 | 2.83 | 61.97 | **+13.52** |
| GPN | 74.68 | 73.23 | 72.55 | 69.60 | 68.91 | 60.41 | 57.76 | 54.82 | 52.58 | 52.58 | 2.46 | 63.81 | **+10.43** |
| ER-GNN | 78.44 | 75.38 | 75.07 | 71.24 | 70.73 | 61.66 | 59.68 | 56.62 | 54.33 | 53.13 | 2.81 | 65.62 | **+9.88** |
| IDLVQ | 77.75 | 78.04 | 77.20 | 74.67 | 74.83 | 66.81 | 65.43 | 62.06 | 60.78 | 58.84 | 2.10 | 69.74 | **+4.18** |
| FACT | **79.43** | **78.95** | 77.01 | 70.39 | 65.47 | 55.24 | 51.04 | 48.90 | 45.74 | 43.70 | 3.97 | 61.58 | **+19.31** |
| EHS | 74.78 | 74.89 | 75.45 | 71.99 | 72.96 | 64.68 | 62.42 | 60.44 | 58.49 | 57.32 | 1.94 | 67.34 | **+5.69** |
| TEEN | 78.65 | 78.02 | 77.14 | 74.05 | 73.15 | 64.01 | 62.02 | 58.64 | 57.31 | 55.88 | 2.53 | 67.88 | **+7.13** |
| HAG-Meta | 70.21 | 70.24 | 68.33 | 66.16 | 64.22 | 62.75 | 62.21 | 62.57 | 61.97 | 60.94 | **1.03** | 64.96 | **+2.07** |
| Geometer | 75.00 | 75.20 | 75.14 | 72.42 | 71.63 | 63.80 | 62.63 | 59.24 | 57.84 | 56.24 | 2.08 | 66.91 | **+6.77** |
| **CAPL(ours)** | 78.77 | 78.08 | **78.85** | **76.59** | **76.03** | **69.26** | **68.01** | **64.87** | **63.28** | **63.01** | 1.75 | **71.68** | |

Table 2: PerformanComparative Results on the three datasets under different N-way K-shot settings. We repeated all experiments 10 times and took the average as the final result. [*]: "Improvement" refers to the increase in accuracy of CAPL compared to the model in the final session.

| Loss functions | | | | | Cora-Full dataset (5-way 5-shot) | | | | | | Amazon-Electronics dataset (10-way 5-shot) | | | | | |
|---|---|---|---|---|---|---|---|---|---|---|---|---|---|---|---|---|
| $\mathcal{L}_n$ | $\mathcal{L}_{inter}$ | $\mathcal{L}_{db}$ | $\mathcal{L}_{kd}$ | $\mathcal{L}_c$ | 0 | 2 | 4 | 6 | 8 | 10 | 0 | 2 | 4 | 6 | 8 | 10 |
| | ✓ | ✓ | ✓ | ✓ | 80.64 | 63.07 | 51.87 | 43.48 | 41.43 | 39.80 | 75.71 | 67.42 | 65.09 | 59.92 | 54.54 | 53.64 |
| ✓ | | ✓ | ✓ | ✓ | **80.71** | 63.21 | 52.10 | 46.33 | 40.96 | 37.92 | 75.31 | 69.16 | 64.00 | 58.31 | 54.21 | 53.84 |
| ✓ | ✓ | | ✓ | ✓ | 80.45 | 57.74 | 44.54 | 40.25 | 37.70 | 35.43 | **76.37** | 64.56 | 61.27 | 56.29 | 52.24 | 51.70 |
| ✓ | ✓ | ✓ | | ✓ | 80.32 | 64.05 | 48.44 | 45.07 | 43.01 | 41.52 | 75.71 | 70.25 | 66.38 | 61.65 | 57.89 | 56.04 |
| ✓ | ✓ | | | ✓ | 80.50 | 61.50 | 47.53 | 40.23 | 36.19 | 32.79 | 76.22 | 64.77 | 58.23 | 54.56 | 50.92 | 49.72 |
| ✓ | ✓ | ✓ | ✓ | ✓ | 80.29 | **64.47** | **52.54** | **47.41** | **44.92** | **42.97** | 75.66 | **70.51** | **68.50** | **62.91** | **60.01** | **59.06** |

Table 3: Ablation study of loss functions comparison on Cora-Full dataset and Amazon-Electronics dataset.

## 4.2 COMPARISON RESULTS

The evaluation process for each dataset is repeated 10 times and the average test results are reported. The experimental results on datasets Cora-Full, Amazon-Electronics, and Amazon-Clothing are shown in Tables 2. From the results, we can see compared to the existing methods, the proposed method demonstrates outstanding general classification performance, which proves the effectiveness of CAPL for GFSCIL. It is clear that our CAPL performs superior to the previous space reservation method IDLVQChen & Lee (2020), EHSDeng & Xiang (2024), and FACTZhou et al. (2022a) during the incremental learning phase, achieving an improvement of over 4% on all three datasets compared to the best-performing methods. This further demonstrates the necessity of class-adaptive space reservation for graph data. In addition, compared to the two existing GFSCIL methods GeometerLu et al. (2022) and HAG-MetaTan et al. (2022), our approach achieved improvements of 3.37%, 4.78%, and 2.07% on the Cora-Full, Amazon-Electronics, and Amazon-Clothing datasets, respectively.

Further, CAPL demonstrates the best performance in terms of average forgetting on Cora-Full and Amazon-Electronics dataset and suboptimal performance on Amazon-Clothing dataset. This indicates that CAPL is more effective in mitigating the catastrophic forgetting problem. Additionally, although our model does not perform optimally in the early stages, it shows superior classification performance compared to other models as the number of classes increases. This demonstrates the effectiveness of the class-adaptive space reservation method for classifying subsequent categories.

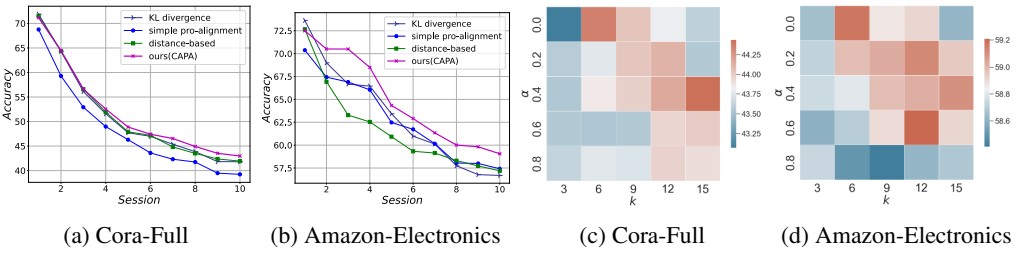

|     (a) Cora-Full     |     (b) Amazon-Electronics     |     (c) Cora-Full     |     (d) Amazon-Electronics     |

Figure 2: Ablation experiment of knowledge distillation methods on Cora-Full (a) and Amazon-Electronics (b) dataset. And parameter analysis of k and $\alpha$ on Cora-Full (c) and Amazon-Electronics (d) dataset. Here we show the accuracy of final session.

## 4.3 ABLATION STUDIES

We do ablation studies in terms of the losses and the knowledge distillation module respectively, on Cora-Full and Amazon-Electronics datasets.

(a) **The losses**: Table 3 presents the classification accuracy on the two datasets with respect to different adoption of losses. We can see each loss is necessary and important to the continuous learning of new classes and the knowledge retention of old classes. Specifically, we can observe that the model performs the worst when neither of the two class-adaptive prototype learning loss, $\mathcal{L}_{db}$ and $\mathcal{L}_{kd}$, are used, which highlights the significance of reserving space for new classes through class-adaptive prototype learning module.

(b) **The knowledge distillation module**: We compare our knowledge distillation module CAPA

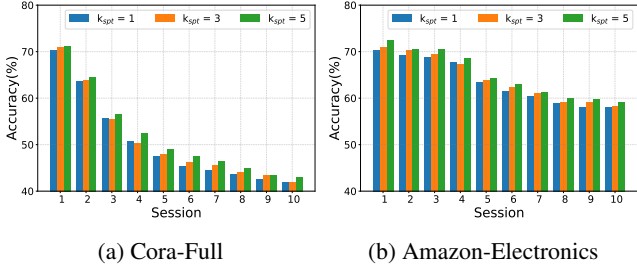

(a) Cora-Full        (b) Amazon-Electronics

Figure 3: Parameter Analysis of support set size of novel classes on Cora-Full and Amazon-Electronics dataset.

with other distillation appraoch, including distance-based knowledge distillation, KL divergence-based knowledge distillation, and simple prototype alignment knowledge distillation. The experimental results are shown in Figure 2a and Figure 2b, which demonstrates the effectiveness of proposed class-adaptive prototype alignment approach by achieving superior classification accuracy and lower forgetting rates.

In this section, we conduct parameter analysis in terms of the number of decay categories $k$ and the decay weight $\alpha$ on Cora-Full dataset and Amazon-Electronics dataset, and the final stage accuracy distribution of each parameter combination is shown in Figure 2c and Figure 2d. Overall, when $k$ is large and $\alpha$ is small, the classification accuracy decreases. This decrease is attributed to the fact that the constraint on prototype representation becomes too weak, leading to the loss of essential classification information for the old classes. In contrast, when $k$ is small or $\alpha$ is large, the final classification accuracy remains relatively low. This suggests that the model's performance is closer to a simple prototype alignment approach, underscoring the critical importance of incorporating class adaptation. This observation further validates the need for class-adaptive prototype alignment approach to achieve improved performance.

Furthermore, we explore the impact of the support set size, $K_{spt}$, on two datasets. By varying the shot number $K_{spt} \in \{1, 3, 5\}$, we observe the resulting differences in model performance. As show in Figure3, we can clearly observe that as the support set size increases, the performance of CAPL improves, indicating that a larger support set size benefits the model's classification ability. At the same time, this demonstrates that CAPL is capable of effectively utilizing a small amount of labeled data for improved classification.

## 5 CONCLUSION

In this paper, we propose a new model CAPL to address the problem of few-shot class incremental learning in graphs. To the best of our knowledge, current works on few-shot class incremental learning treat all classes equally, which is not suitable for non-i.i.d. graph data. Our proposed CAPL adopts different strategies for different classes based on their relationships. On one hand, it achieves class-adaptive prototype clustering based on the positional relationships among classes to reserve space for newly arrived classes. On the other hand, it proposes a class-adaptive prototype alignment method based on the relationships between new and old classes to realize knowledge distillation. Extensive experiments on three commonly used graph datasets demonstrate that CAPL effectively outperforms state-of-the-art performance.

## 6 LIMITATIONS AND FUTURE WORK

One limitation of our work is that it only studies the performance of FSCIL on homogeneous graphs and does not extend to heterogeneous graphs, which remains an unexplored and challenging area for future research. In addition, although our approach effectively alleviate catastrophic forgetting, the accuracy of old categories still decreases, requiring further research for a more effective solution.

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

# A RELATED WORK

## A.1 CLASS-INCREMENTAL LEARNING

The goal of class incremental learning (CIL) is to learn a model that can continuously recognize new classes without forgetting old ones. Storing class examples from old sessions and replaying them in new sessions is a straightforward strategy that can help the model maintain good performance on old classes Isele & Cosgun (2018); Liu et al. (2021b); Rebuffi et al. (2017); Rolnick et al. (2019). Furthermore, knowledge distillation transfers knowledge from a teacher model to a student model, enabling the student model to compress and distill while retaining the rich knowledge of the teacher model Gou et al. (2021); Hinton et al. (2015); Kang et al. (2022); Rebuffi et al. (2017). iCaRL Rebuffi et al. (2017) uses knowledge distillation and replays old class exemplars while learning new classes to prevent catastrophic forgetting. CLAD Xu et al. (2024) reduces forgetting effectively by predicting which old classes are most likely to be forgotten and by regularizing the separation of representations between old and new classes during training on new classes. Graph incremental learning can be categorized into three branches: regularization Liu et al. (2021a), replay Zhang et al. (2022b); Zhou & Cao (2021), and architecture-based methods Zhang et al. (2022a). The TWP method Liu et al. (2021a) uses regularization to retain the topological structure of historical graphs. HPN Liu et al. (2021a) introduces a redesigned three-layer prototype architecture to enhance representation learning. ERGNN Zhou & Cao (2021) incorporates memory replay by storing key representative nodes, while SSM Zhang et al. (2022b) preserves structural information by storing sparse subgraphs in a dedicated repository. However, these methods failed to achieve the expected performance in few-shot scenarios.

## A.2 GRAPH FEW-SHOT LEARNING

In the real world, class data is not always abundant. When data is scarce, the model needs to learn as much as possible about the classes under conditions of limited samples without encountering overfitting problems. To address this challenge, few-shot learning (FSL) aims to train a model that can accurately classify new classes with very limited training samples. Meta-GNN Zhou et al. (2019) is the first to introduce the meta-learning paradigm into GNNs Cao et al. (2016); Defferrard et al. (2016); Hamilton et al. (2017); Veličković et al. (2017); Xu et al. (2018) to address FSL problems on graphs. G-META Huang & Zitnik (2020) utilizes local subgraphs for graph meta-learning to acquire transferable knowledge. In addition, GPN Ding et al. (2020) solves the problem of few-shot node classification in attribute networks by employing prototype networks in GNNs. GFL Yao et al. (2020) leverages prior knowledge gained from auxiliary graphs to facilitate the transfer of this knowledge to a new target graph. Although these methods perform well in few-shot tasks, they struggle to adapt to the CIL scenario.

## A.3 GRAPH FEW-SHOT CLASS-INCREMENTAL LEARNING

As a variant of CIL, FSCIL involves only a small number of new classes and training data at each incremental learning stage. This means it not only faces the catastrophic forgetting problem inherent in CIL but also the risk of overfitting due to the insufficient number of samples. Some works Chen & Lee (2020); Deng & Xiang (2024); Zhou et al. (2022a) have improved the model's compatibility and performance in handling new classes by reserving embedding space for them. Recently, TEEN Wang et al. (2024) proposes a prototype calibration strategy that enhances new class discriminability by blending new and weighted base class prototypes. Although these methods perform well on image Tao et al. (2020); Kukleva et al. (2021); Liu et al. (2022); Zhou et al. (2022b); Zhao et al. (2023); Smith et al. (2024) and natural language processing domains Qin & Joty (2021; 2022); Wang et al. (2022), they fail to fully capture the rich information inherent in graph-structured data. Hence, the development of FSCIL methods tailored for graph-structured data is crucial. HAG-Meta Tan et al. (2022) first proposed the GFSCIL problem and addressed the issue of the class imbalance with a hierarchical attention mechanism. Geometer Lu et al. (2022) leverages knowledge distillation and geometric metric learning to learn prototype representations, mitigating the issue of prototype overlap and forgetting. However, both of these methods do not consider the edge connections between nodes and treat all classes equally. In contrast, our proposed CAPL adaptively handles each class based on the relationships between classes.

# B DETAILS REGARDING THE COMPARATIVE EXPERIMENTS

## B.1 DETAILS OF THE DATASET

- Cora-Full Bojchevski & Günnemann (2017) is a well-known citation network based on paper topic labels. In this dataset, each paper is considered as a node, and edges are formed between papers based on their citation relationships.It holds the largest network scale and the highest number of categories among the academic networks known to us.

- Amazon-Clothing McAuley et al. (2015) is a product network built with the products in "Clothing, Shoes and Jewelry" on Amazon. In this dataset, each product is considered as a node, and its description is used to construct the node attributes. We use the substitutable relationship ("also viewed") to create links between products.

- Amazon-Electronics McAuley et al. (2015) is another Amazon product network which contains products belonging to "Electronics". Similar to the first dataset, each node denotes a product and its attributes represent the product description. Note that here we use the complementary relationship ("bought together") between products to create the edges.

## B.2 DETAILS OF BASELINE

- PN Snell et al. (2017): Prototype Network firstly proposed for few-shot image classification. We adopt the key idea and implement PN for node classification. In the experiment, the convolutional network was replaced with a GAT and then applied to GFSCIL.

- GPN Ding et al. (2020): GPN effectively addresses the problem of few-shot node classification on attributed networks by introducing prototypes in graph neural networks to represent each class. It leverages graph neural networks to extract node feature representations and designs a node evaluator to learn the informativeness of each labeled node, thereby obtaining more representative class prototypes. Additionally, GPN employs the meta-learning paradigm to train the model for few-shot scenarios, enhancing its generalization ability. During classification, GPN uses the Euclidean distance from sample representations to class prototypes for classification.

- ER-GNN Zhou & Cao (2021): ERGNN proposes a continual graph learning paradigm, utilizing an experience replay mechanism to store knowledge from previous tasks as experience nodes and replay these experience nodes when learning new tasks to effectively mitigate the catastrophic forgetting problem in graph neural networks during continual learning tasks. Additionally, ERGNN introduces three experience node selection strategies, including mean of feature, coverage maximization, and influence maximization. For a fair comparison with CAPL, we extend ERGNN to the few-shot learning scenario.

- Geometer Lu et al. (2022): Geometer learns prototype representations for each class through an attention mechanism and employs geometric metric learning, including three geometric loss functions: intra-class proximity, inter-class consistency, and inter-class separability to optimize the prototype representations. Additionally, the model introduces a teacher-student knowledge distillation method to prevent forgetting old classes and uses a biased sampling strategy to alleviate class imbalance issues.

- HAG-Meta Tan et al. (2022): HAG-Meta introduces few-shot class-incremental learning to the graph domain for the first time and proposes the problem of graph few-shot class-incremental learning. HAG-Meta utilizes a graph pseudo incremental learning paradigm to simulate real incremental scenarios, providing the model with a rich training environment to enhance its adaptability. Meanwhile, the hierarchical attention mechanism dynamically adjusts the contribution of different tasks to model training, effectively balancing the model's learning on old and new classes. This effectively addresses the class imbalance problem faced by graph neural networks when handling incremental learning tasks.

- IDLVQ Chen & Lee (2020): IDLVQ leverages learning vector quantization in the deep embedding space to represent knowledge and learns reference vectors for new classes with a few samples while updating the feature extractor to achieve incremental learning. Additionally, IDLVQ introduces intra-class variation regularization, less forgetting regularization, and a reference vector calibration mechanism to mitigate the forgetting of old class

knowledge. Moreover, IDLVQ employs a boundary-based loss function to avoid unnecessary model parameter updates.

- FACT Zhou et al. (2022a): During the pre-training phase, FACT allocates multiple virtual prototypes in the embedding space to reserve space for future new classes. These virtual prototypes help to promote tighter clustering of instances of the same class, thereby reserving more space for new classes. Additionally, the method generates virtual instances through instance mixing to simulate potential future class distributions, enabling the model to adapt to future classes. During the inference phase, these virtual prototypes act as informative basis vectors, aiding in the construction of a more robust incremental classifier.

- EHS Deng & Xiang (2024): EHS utilizes the odd-symmetric activation function tanh to ensure the integrity and uniformity of the embedding space, allocating different regions within the embedding space for base classes and future classes, thereby increasing the distance between class distributions. Additionally, EHS employs virtual instances to enable the model to predict potential future samples. Consequently, EHS effectively mitigates the issue of insufficient feature space in few-shot class-incremental learning.

- TEEN Wang et al. (2024): TEEN identifies that existing methods often misclassify new class samples as similar base class samples, resulting in poor performance for new classes. By analyzing the training process of the feature extractor, it is found that training only with base class samples can still effectively represent the semantic similarity between base and new classes. Based on this, TEEN proposes a training-free prototype calibration strategy that enhances the separability of new classes by integrating new class prototype with weighted base class prototype.

In the experiment, for methods in the image domain, we replaced the feature extractor with a two-layer GAT network. Additionally, the settings for all methods in the experiment remained consistent.

