# OpenReview forum: "CAPL: Graph Few-Shot Class-Incremental Learning Via Class-Adaptive Prototype Learning"
_ICLR.cc/2026/Conference — Submitted to ICLR 2026_

### Official Review · Reviewer_SHiP · 2025-10-26

**Soundness:** 2
**Presentation:** 2
**Contribution:** 2
**Rating:** 2
**Confidence:** 3

**Summary:**

This paper tackles the challenges of Graph Few-Shot Class-Incremental Learning (GFSCIL), including catastrophic forgetting and inter-class confusion. The authors propose a Class-Adaptive Prototype Learning (CAPL) framework that adaptively adjusts class prototypes based on inter-class relationships. It consists of a spatial reservation module to allocate space for new classes and a prototype alignment module to balance knowledge retention and adaptation. Experiments show that CAPL effectively mitigates forgetting and outperforms existing methods.

**Strengths:**

The paper proposes a clear and effective framework for Graph Few-Shot Class-Incremental Learning (GFSCIL). The Class-Adaptive Prototype Learning (CAPL) method adaptively handles different classes, reducing confusion between new and old classes and alleviating forgetting. The design is intuitive, well-motivated, and achieves strong performance improvements over existing methods.

**Weaknesses:**

The paper lacks clarity on whether base data are accessible during incremental learning. Comparisons with related GNN-based methods are missing. Graph structures are not fully utilized beyond prototype computation, and no t-SNE visualizations are provided to show feature space evolution.

**Questions:**

1. In the incremental stage, is the data from the base stage still accessible? If it is available, will forgetting of the base classes still occur? In that case, does it still make sense to suppress the forgetting of base classes?

2. Currently, there already exist several methods that employ Graph Neural Networks (GNNs) for Few-Shot Class-Incremental Learning (FSCIL), such as CEC [1], as well as approaches that conduct experiments on graph-structured datasets, such as SMILE [2]. Therefore, this paper should include comparative experiments with these types of methods.

3. There are still some spelling errors in this paper. For example, in Figure 1, the word “Attention” in the base stage section is misspelled as “Atention.”

4. In this paper, the graph properties are only utilized when computing prototypes. In the subsequent process, however, the edge-weight relationships between nodes in the graph are not exploited, and the connections among instances are not fully explored. Therefore, introducing a graph structure into FSCIL appears to be non-essential in this context.

5. The proposed method mainly focuses on adjusting and optimizing the feature space. However, the experimental section does not include any t-SNE visualizations of the feature space. It is recommended to add t-SNE plots that illustrate the changes in the feature distribution and the positions of class prototypes.

[1] Zhang C, Song N, Lin G, et al. Few-shot incremental learning with continually evolved classifiers. In Proceedings of the IEEE/CVF conference on computer vision and pattern recognition. 2021: 12455-12464.

[2] Liu Y, Li M, Giunchiglia F, et al. Dual-level Mixup for Graph Few-shot Learning with Fewer Tasks. In Proceedings of the ACM on Web Conference 2025. 2025: 2646-2656.

---

### Official Review · Reviewer_KYGh · 2025-10-29

**Soundness:** 2
**Presentation:** 2
**Contribution:** 2
**Rating:** 4
**Confidence:** 4

**Summary:**

This work proposes CAPL, a class-adaptive prototype learning framework for graph few-shot class-incremental learning (GFSCIL). It tackles catastrophic forgetting, overfitting, and class confusion by introducing two key modules: Class-Adaptive Prototype Alignment (CAPA), which performs flexible knowledge distillation, and Class-Adaptive Prototype Clustering (CAPC), which reserves feature space adaptively for different classes. Experiments on Cora-Full, Amazon-Electronics, and Amazon-Clothing show that CAPL outperforms prior methods by over 3% on average accuracy.

**Strengths:**

1. The work tries to address an interesting problem—few-shot class-incremental learning on graphs—where data scarcity and structural dependencies make standard FSCIL methods ineffective.
2. The proposed class-adaptive framework is technically sound, and ablation studies verify the contribution of its main components.

**Weaknesses:**

1. Lack of theoretical analysis or interpretability case for CAPL. While the model is decomposed into three modules (prototype learning, CAPC, CAPA), their claimed effects (e.g., “CAPC further improves intra-class compactness and reserves space for new classes”) are not convincingly supported by theoretical justification or targeted empirical analysis. The paper mainly relies on overall accuracy gains rather than directly validating the intended mechanisms.
2. Incremental novelty. The main ideas—prototype-based classification, clustering regularization, and class-aware distillation—are largely adapted from existing FSCIL and GNN literature. The“class-adaptive” design appears incremental rather than conceptually transformative.
3. Writing and clarity. While the paper presents valuable ideas, some sections, particularly the introduction and methods, could benefit from more concise and clearer phrasing. A more streamlined presentation and a more explicit motivation for each component would enhance the readability and overall clarity of the manuscript.

**Questions:**

1. The paper lists four challenges of graph few-shot class-incremental learning, but it is unclear which specific challenge CAPL primarily aims to solve. The authors should clarify whether the method mainly targets catastrophic forgetting, feature overlap, or another issue, rather than broadly claiming to address all four.

2. How exactly does the CAPC module influence the geometry of the feature space? Could the authors provide quantitative or visual evidence (e.g., t-SNE, intra-/inter-class distances) to show that it indeed improves intra-class compactness and inter-class separability?

3. The definitions and roles of the base stage and incremental learning stage are not clearly explained. A clearer description of their differences, objectives, and how they interact within CAPL would help readers follow the training process.

---

### Official Review · Reviewer_Ce4A · 2025-10-31

**Soundness:** 1
**Presentation:** 2
**Contribution:** 2
**Rating:** 2
**Confidence:** 4

**Summary:**

This paper addresses the problem of Graph Few-Shot Class-Incremental Learning (GFSCIL) by proposing a Class-Adaptive Prototype Learning (CAPL) method. The authors argue that existing approaches treat all classes equally and do not consider inter-class relationships.  The proposed method is evaluated on three graph datasets and shows improvements over baselines in terms of average accuracy and forgetting metrics.

**Strengths:**

1. The experimental evaluation includes multiple datasets and detailed ablation studies that demonstrate the contribution of different components.
2. The paper provides motivation for why class-adaptive approaches are needed for graph data, emphasizing the non-i.i.d. nature of graph nodes and complex inter-class relationships.

**Weaknesses:**

1 The core ideas presented are largely combinations or incremental modifications of well-established techniques in the (FS)CIL domain. The method is built on a foundation of standard prototype learning [1] and uses prototype alignment via knowledge distillation, which is a common technique for mitigating catastrophic forgetting in continual learning [2, 3, 4]. Similarly, the "class-adaptive" knowledge distillation (CAPA) is a distance-based weighting scheme for a standard KD loss.

2 The proposed method also appears to lack component integration, feeling more like a collection of disparate loss terms rather than a unified, principled framework. There is no clear theoretical or practical explanation of how the "space reservation" achieved by CAPC interacts with or benefits the "prototype flexibility" provided by CAPA.

3 In Equation 7 ,while borrowed from clustering, its application here is not well-justified. The DB index assumes symmetric relationships between clusters, but the paper emphasizes directed, asymmetric relationships in graphs. The max operation focuses on one neighbouring class, but graphs often have multiple important neighbours.

4 The product in Equation 10 can make weights extremely small when many new classes exist, potentially causing numerical instability.

5 More experiments should be conducted on commonly used FSCIL datasets (e.g. CUB, Mini-ImageNet, CIFAR100) to validate the effectiveness of the method.

6 No computational cost analysis is provided. Class-adaptive processing likely increases complexity significantly


[1] Few-Shot Incremental Learning with Continually Evolved Classifiers

[2] Semantic-aware Knowledge Distillation for Few-Shot Class-Incremental Learning

[3] SSFE-Net: Self-Supervised Feature Enhancement for Ultra-Fine-Grained Few-Shot Class Incremental Learning

[4] Few-Shot Class-Incremental Learning via Relation Knowledge Distillation

**Questions:**

Please see the weaknesses.

---

### Official Review · Reviewer_UPv6 · 2025-10-31

**Soundness:** 2
**Presentation:** 3
**Contribution:** 1
**Rating:** 2
**Confidence:** 5

**Summary:**

This paper tackles graph few-shot class-incremental learning (FSCIL) problem and proposes class-adaptive prototype learning.
The authors propose several loss functions including $\mathcal{L}_c$, $\mathcal{L}_n$, $\mathcal{L}inter$, $\mathcal{L}db$, and $\mathcal{L}kd$ that encourage learning representation and class-prototypes that exhibit compact intra-class clusters and discriminative inter-class prototypes.
The experimental results show that the proposed loss functions contribute to the performance improvement.

**Strengths:**

The experimental results show that the proposed method outperforms the existing FSCIL methods.

**Weaknesses:**

$\textbf{1. Lack of Novelty}$

The motivation to form compact and discriminative class clusters to reserve feature space for new classes has already been addressed by many previous works in FSCIL [1,2,3,4].
The only difference appears to be in the loss function design, but the reviewer finds that the difference is minor and the proposed loss functions seem to be hand-crafted and less insightful.

$\textbf{2. Lack of comparison}$

Some previous [5,6] works show that reducing intra-class distance and maximizing the inter-class margin is not beneficial in the context of FSCIL, because this might hurt the transferability of learned representation during the base session.
To make this paper more appealing, it appears to be necessary to present more discussion and analysis of the proposed method compared to the previous representation learning based FSCIL methods [1,2,3,4,5,6].

$\textbf{3. Inaccurate notations or mathematical expressions}$

In L111, The equation is incomplete: $\Delta \mathcal{C}_i \cap \Delta \mathcal{C}_j = $.
In L152, the equation for $p_i$ is inaccurate: $z_i$ is not defined and the subscript should be $j$ not $i$.
In L156, the equation for $\boldsymbol{Z}_i$ is confusing. What does $||$ indicate? Does it mean $\text{Concatenate}( \[\boldsymbol{z}_j | \boldsymbol{v}_j \in S_t^j\])$?
In Equation 3, $C$ should be $\mathcal{C}$.
These inaccurate expressions make the paper difficult to follow and unprofessional.

[1] Hersche etal, "Constrained few-shot class-incremental learning", in CVPR 2022

[2] Song etal, "Learning with fantasy: semantic-aware virtual contrastive constraint for few-shot class-incremental learning" in CVPR 2023

[3] Yang etal, "Neural collapse inspired feature-classifier alignment for few-shot class incremental learning", in ICLR 2023

[4] Zhou etal, "Forward compatible few-shot class-incremental learning" in CVPR 2022

[5] Zou etal, "Margin-Based Few-Shot Class-Incremental Learning with Class-Level Overfitting Mitigation", in NeurIPS 2022

[6] Oh etal, "CLOSER: Towards Better Representation Learning for Few-Shot Class-Incremental Learning", in ECCV 2024

**Questions:**

1. During the incremental stage, if the model parameters are updated, the prototypes for old classes might become outdated. Although the authors propose adjusting prototypes for old classes, it is not guaranteed that the adjusted prototypes effectively cover the actual feature shift for old classes.

2. It seems that the proposed method can be applied to the general FSCIL problem like image classification. Since many FSCIL works evaluate their methods on image classification task, it would be better to compare the proposed method with them in the commonly used setup.

---

### Meta-Review · Area_Chair_mFWa · 2025-12-26

**Summary:**

This paper proposes a CAPL method for knowledge increment with few samples, addressing the problem of continuously increasing classes on a graph. The authors propose a class-adaptive spatial reservation module to allocate larger spaces for the arrival of new classes. Experiments demonstrate that this strategy effectively mitigates inter-class confusion caused by class increments.

**Reviewer Concerns:**

During the peer review process, reviewer UPv6 expressed a lack of innovation, noting numerous adjustments and learnings regarding intra- and inter-class relationships and associations in FSCIL. This included a lack of necessary comparative experiments, as detailed in reviewer UPv6's comments. Reviewer Ce4A also raised discussions about the core ideas, suggesting the authors focused more on a set of multiple loss distributions.

**Reviewer Scores:**

This paper received three clear rejections (2 points each) and one negative rejection (4 points). This indicates a consensus among the reviewers.

Based on the above review comments and scores, the Area Chair (AC) agreed, and the authors did not submit any response or discussion, including the final statement. The AC believes the paper still requires some revisions.

---

### Decision · Program_Chairs · 2026-01-26

Reject